# OMNIINPUT: A MODEL-CENTRIC EVALUATION FRAMEWORK THROUGH OUTPUT DISTRIBUTION

## ABSTRACT

We propose a novel model-centric evaluation framework, OMNIINPUT, to evaluate the quality of an AI/ML model's predictions on all possible inputs (including human-unrecognizable ones), which is crucial for AI safety and reliability. Unlike traditional data-centric evaluation based on pre-defined test sets, the test set in OMNIINPUT is self-constructed by the model itself and the model quality is evaluated by investigating its output distribution. We employ an efficient sampler to obtain representative inputs and the output distribution of the trained model, which, after selective annotation, can be used to estimate the model's precision and recall at different output values and a comprehensive precision-recall curve. Our experiments demonstrate that OMNIINPUT enables a more fine-grained comparison between models, especially when their performance is almost the same on pre-defined datasets, leading to new findings and insights for how to train more robust, generalizable models.

## 1 INTRODUCTION

A safe, reliable AI/ML model deployed in real world should be able to make reasonable predictions on all the possible inputs, including uninformative ones. For instance, an autonomous vehicle image processing system might encounter carefully designed backdoor attack patterns (that may look like noise) (Li et al., 2022; Liu et al., 2020b), which can potentially lead to catastrophic accidents if such backdoor patterns interfere the stop sign or traffic light classification.

Existing evaluation frameworks are mostly, if not all, *data-centric*, meaning that they are based on pre-defined, annotated datasets. The drawback is the lack of a comprehensive understanding of the model's fundamental behaviors over *all* possible inputs. Recent literature showed that a great performance on a pre-defined (in-distribution) test set cannot guarantee a strong generalization to different regions in the input space, such as out-of-distribution (OOD) test sets (Liu et al., 2020a; Hendrycks & Gimpel, 2016; Hendrycks et al., 2019; Hsu et al., 2020; Lee et al., 2017; 2018) and adversarial test sets (Szegedy et al., 2013; Rozsa et al., 2016; Miyato et al., 2018; Kurakin et al., 2016). One possible reason for poor generalization in the open-world setting is overconfident prediction (Nguyen et al., 2015), where the model could wrongly predict OOD input as in-distribution objects with high confidence.

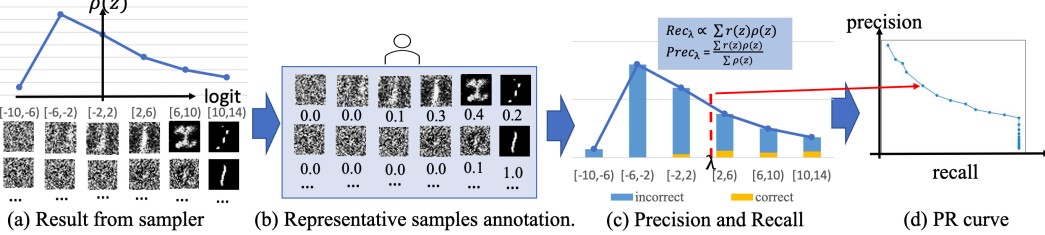

Figure 1: An overview of our novel OMNIINPUT evaluation framework. (a) Use an efficient sampler, e.g. GWL (Liu et al., 2023), to obtain the output distribution $\rho(z)$ and sample representative inputs; (b) Annotate representative inputs; (c) Estimate the precision and recall at different threshold $\lambda$. $r(z)$ denotes the precision of the model within the bin of output value $z$; (d) Construct a precision-recall curve as evaluation results.

Inspired by the evaluation frameworks for generative models (Heusel et al., 2017; Salimans et al., 2016; Naeem et al., 2020; Sajjadi et al., 2018; Cheema & Urner, 2023), we propose a novel model evaluation approach from a *model-centric* perspective: after the model is trained, we construct the test set from the model's self-generated, representative inputs corresponding to different model output values. We then annotate these samples, and estimate the model performance over the entire input space using the model's output distribution. While existing generative model evaluation frameworks are also model-centric, we are the first to leverage the output distribution as a unique quantity to generalize model evaluation from representative inputs to the entire input space. To illustrate our proposed novel evaluation framework OMNIINPUT, we focus on a binary classification task of classifying if a picture is digit 1 or not. As shown in Fig. 1, it consists of four steps:

(a) We employ a recently proposed sampler to obtain the output distribution $\rho(z)$ of the trained model (where $z$ denotes the output value of the model) over the entire input space (Liu et al., 2023) and efficiently sample representative inputs from different output value (e.g., logit) bins. The output distribution is a histogram counting the number of inputs that lead to the same model output. In the open-world setting without any prior knowledge of the samples, all possible inputs should appear equally.

(b) We annotate the sampled representative inputs to finalize the test set, e.g., rate how likely the picture is digit 1 using a score from 0 to 1.[1]

(c) We compute the precision for each bin as $r(z)$, then estimate the precision and recall at different threshold values $\lambda$. When aggregating the precision across different bins, a weighted average of $r(z)$ by the output distribution $\rho(z)$ is required i.e., $\frac{\sum_{z \geq \lambda} r(z) \cdot \rho(z)}{\sum_{z \geq \lambda} \rho(z)}$. See Sec. 2.2 for details.

(d) We finally put together the precision-recall curve for a comprehensive evaluation of the model performance over the entire input space.

OMNIINPUT samples the representative inputs solely by the model itself, eliminating possible human biases introduced by the test data collection process. The resulting precision-recall curve can help decide the limit of the model in real-world deployment. The overconfident prediction issue can also be quantified precisely manifested by a low precision when the threshold $\lambda$ is high.

Our OMNIINPUT framework enables a more fine-grained comparison between models, especially when their performance is almost the same on the pre-defined datasets. Take the MNIST dataset as an example, many models (e.g., ResNet, CNN, and multi-layer Perceptron network (MLP)) trained by different methods (e.g., textbook cross-entropy (CE), CE with (uniform noise) data augmentation, and energy-based generative framework) can all achieve very high or nearly perfect performance. Our experiments using OMNIINPUT reveals, for the first time, the differences in the precision-recall curves of these models over the entire input space and provides new insights. They include:

- The architectural difference in MLP and CNN, when training with the CE loss and original training set, can lead to significant difference in precision-recall curves. CNN prefers images with dark background as representative inputs of digit 1, while MLP prefers to invert the background of zeros as digit 1.
- Different training schemes used on the same ResNet architecture can lead to different performance. Adding noise to the training set in general can lead to significant improvements in precision and recall than using energy-based generative models; however, the latter leads to samples with a better visual diversity. These results suggest that combining the generative and classification objectives may be the key for the model to learn robust classification criteria for all possible samples.

Additionally, we have evaluated DistilBERT for sentiment classification and ResNet on CIFAR (binary classification) using OMNIINPUT. Our results indicate a significant number of overconfident predictions, a strong suggestion of poor performance in the entire input space. It is worth mentioning that these findings are specific to the models we trained. Thus, this is not a conclusive study of the differences of the models with different training methods and architectures, but a demonstration of how to use our OMNIINPUT framework to quantify the performance of the models and generate new insights for future research. The contributions of this work are as follows:

---

[1]In data-centric evaluations, the pre-defined test set is typically human-annotated as well. Our experiments show that 40 to 50 human annotations per output bin are enough for a converged precision-recall curve (Fig. 4), hence human involvement required is significantly smaller in our method.

- We propose to evaluate AI/ML models by considering all the possible inputs with equal probability, which is crucial to AI safety and reliability.
- We develop a novel model-centric evaluation framework, OMNIINPUT, that constructs the test set by representative inputs, and leverages output distribution to generalize the evaluation assessment from representative inputs to the entire input space. This approach largely eliminates the potential human biases in the test data collection process and allows for a comprehensive understanding and quantification of the model performance.
- We apply OMNIINPUT to evaluate various popular models paired with different training methods. The results reveal new findings and insights for how to train robust, generalizable models.

## 2 THE OMNIINPUT FRAMEWORK

In this section, we present a detailed background on sampling the output distribution across the entire input space. We then propose a novel model-centric evaluation framework OMNIINPUT in which we derive the performance metrics of a neural network (binary classifier) from its output distribution.

### 2.1 OUTPUT DISTRIBUTION AND SAMPLER

**Output Distribution.** We denote a trained binary neural classifier parameterized by $\theta$ as $f_\theta : \mathbf{x} \to z$ where $\mathbf{x} \in \Omega_T$ is the training set, $\Omega_T \subseteq \{0, ..., N\}^D$, and $z \in \mathbb{R}$ is the output of the model. In our framework, $z$ represents the logit and each of the $D$ pixels takes one of the $N + 1$ values.

The output distribution represents the frequency count of each output logit $z$ given the entire input space $\Omega = \{0, ..., N\}^D$. In our framework, following the principle of equal *a priori* probabilities, we assume that each input sample within $\Omega$ follows a uniform distribution. This assumption is based on the notion that every sample in the entire input space holds equal importance for the evaluation of the model. Mathematically, the output distribution, denoted by $\rho(z)$, is defined as:

$$\rho(z) = \sum_{\mathbf{x} \in \Omega} \delta(z - f_\theta(\mathbf{x})),$$

where $\delta$ is the Dirac delta function.

**Samplers** The sampling of an output distribution finds its roots in physics, particularly in the context of the sampling of the density of states (DOS) (Wang & Landau, 2001; Vogel et al., 2013; Cunha-Netto et al., 2008; Junghans et al., 2014; Li & Eisenbach, 2017; Zhou et al., 2006), but its connection to ML is revealed only recently (Liu et al., 2023).

The **Wang–Landau (WL) algorithm** Wang & Landau (2001) aims to sample the output distribution $\rho(z)$ which is unknown in advance. In practical implementations, the "entropy" (of discretized bins of $z$), $\tilde{S}(z) = \log \tilde{\rho}(z)$, is used to store the instantaneous estimation of the ground truth $S(z) = \log \rho(z)$. The WL algorithm leverages the reweighting technique, where the sampling weight $w(\mathbf{x})$ is inversely proportional to the instantaneous estimation of the output distribution:

$$w(\mathbf{x}) \propto \frac{1}{\tilde{\rho}(f_\theta(\mathbf{x}))}. \tag{1}$$

When the approximation $\tilde{\rho}(z)$ converges to the true value $\rho(z)$, the entire output space would be sampled uniformly.

The fundamental connection between the output distribution of neural networks and the DOS in physics has been discovered and elucidated in Ref. (Liu et al., 2023). Additionally, it is shown that the traditional Wang–Landau algorithm sometimes struggles to explore the parameter space if the MC proposals are not designed carefully. **Gradient Wang–Landau sampler (GWL)** (Liu et al., 2023) circumvent this problem by incorporating a gradient MC proposal similar to GWG (Grathwohl et al., 2021), which improves Gibbs sampling by picking the pixels that are likely to change. The GWL sampler has demonstrated the feasibility and efficiency of sampling the entire input space for neural networks.

The key component of the output distribution samplers is that they can sample the output space equally and efficiently, thereby providing a survey of the input-output mapping for *all* the possible logits. This is in contrast with traditional MCMC samplers which are *biased* to sample the logits corresponding to high log-likelihood (possible informative samples) over logits correspond to low log-likelihood (noisy and uninformative samples).

## 2.2 MODEL-CENTRIC EVALUATION

Our model evaluation framework revolves around the output distribution sampler. Initially, we obtain the output distribution and the representative inputs exhibiting similar output logit values.

**Representative Inputs.** Although there are exponentially many uninformative samples in the entire input space, it is a common practice in generative model evaluation to generate (representative) samples by sampling algorithms and then evaluate samples, such as Fréchet Inception Distance (FID) (Heusel et al., 2017). In our framework, other sampling algorithms can also be used to collect representative inputs. There should be no distributional difference in the representative inputs between different samplers (Fig. 8). However, Wang–Landau type algorithms provide a more effective means for traversing across the logit space and are hence more efficient than traditional MCMC algorithms in sampling the representative inputs from the output distribution.

**Normalized Output Distribution.** To facilitate a meaningful comparison of different models based on their output distribution, it is important to sample the output distribution of (all) possible output values to ensure the *normalization* can be calculated as accurately as possible. We leverage the fact that the entire input space contains an identical count of $(N + 1)^D$ samples for all models under comparison Landau et al. (2004). Consequently, the normalized output distribution $\rho(z)$ can be expressed as:

$$\log \rho(z) = \log \hat{\rho}(z) - \log \sum_z \hat{\rho}(z),$$

where $\hat{\rho}(z)$ denotes the unnormalized output distribution.

**Annotation of Samples.** For our classifiers, we designate a specific class as the target class. The (human) evaluators would assign a score to each sample within the same "bin" of the output distribution (each "bin" collects the samples with a small range of logit values $[z - \Delta z, z + \Delta z)$). This score ranges from 0 when the sample completely deviates from the evaluator's judgment for the target class, to 1 when the sample perfectly aligns with the evaluator's judgment. Following the evaluation, the average score for each bin, termed "precision per bin", $r(z)$, is calculated. It is the proportion of the total evaluation score on the samples relative to the total number of samples within that bin. We have 200-600 bins for the experiments.

**Precision and Recall.** Without loss of generality, we assume that the target class corresponds to large logit values: we define a threshold $\lambda$ such that any samples with $z \geq \lambda$ are predicted as the target class. Thus, the precision given $\lambda$ is defined as

$$\text{precision}_\lambda = \frac{\sum_{z \geq \lambda}^{+\infty} r(z)\rho(z)}{\sum_{z \geq \lambda}^{+\infty} \rho(z)}.$$

The numerator is the *true positive* and the denominator is the sum of true positive and false positive. This denominator can be interpreted as the *area under curve* (AUC) of the output distribution from the threshold $\lambda$ to infinity.

When considering recall, we need to compute the total number of ground truth samples that the evaluators labeled as the target class. This total number of ground truth samples remains constant (albeit unknown) over the entire input space. Hence recall is proportional to $\sum_{z \geq \lambda}^{+\infty} r(z)\rho(z)$:

$$\text{recall}_\lambda = \frac{\sum_{z \geq \lambda}^{+\infty} r(z)\rho(z)}{\text{number of positive samples}} \propto \sum_{z \geq \lambda}^{+\infty} r(z)\rho(z).$$

A higher recall indicates a better model. As demonstrated above, the output distribution provides valuable information for deriving both precision and (unnormalized) recall. These metrics can be utilized for model evaluation through the precision-recall curve, by varying the threshold $\lambda$. In the extreme case where $\rho(z)$ differs significantly for different $z$, $\text{precision}_\lambda$ is approximated as $r(z^*)$ where $z^* = \arg\max_{z \geq \lambda} \rho(z)$ and $\text{recall}_\lambda$ is approximated as $\max_{z \geq \lambda} r(z)\rho(z)$.

**Quantifying Overconfident Predictions in OMNIINPUT.** Overconfident predictions refer to the samples that (a) the model predicts as positive with very high confidence (i.e., above a very high threshold $\lambda$) but (b) human believes as negative. The ratio of overconfident predictions over the total positive predictions is simply $1 - \text{precision}_\lambda$ in OMNIINPUT. Moreover, even if two models have

nearly the same (high) precision, the difference in (unnormalized) recall $\text{recall}_\lambda$ can indicate which model captures more ground-truth-positive samples. Therefore, compared to methods that only quantify overconfident prediction, OMNIINPUT can offer a deeper insight of model performance using recall.

**Scalability.** Our OMNIINPUT framework mainly focuses on how to leverage the output distribution for model evaluation over the entire input space. To handle larger input spaces and/or more complicated models, more efficient and scalable samplers are required. However, it is beyond the scope of this paper and we leave it as a future work. Our evaluation framework is parallel to the development of the samplers and will be easily compatible to new samplers.

## 3   EXPERIMENTS ON MNIST AND RELATED DATASETS

The entire input space considered in our experiment contains $256^{28 \times 28}$ samples (i.e., $28 \times 28$ gray images), which is significantly larger than any of the pre-defined datasets, and even larger than the number of atoms in the universe (which is about $10^{81}$).

**Models for Evaluation.** We evaluate several backbone models: convolution neural network (**CNN**), multi-layer Perceptron network (**MLP**), and **ResNet** (He et al., 2015). The details of the model architectures are provided in Appendix A. We use the MNIST training set to build the classifiers, but we extract only the samples with labels $\{0, 1\}$, which we refer to as **MNIST-0/1**. For generative models, we select only the samples with **label=1** as **MNIST-1**; samples with labels other than **label=1** are considered OOD samples. We build models using different training methods: (1) Using the vanilla binary cross-entropy loss, we built **CNN-MNIST-0/1** and **MLP-MNIST-0/1**[2] which achieve test accuracy of 97.87% and 99.95%, respectively; (2) Using the binary cross-entropy loss and data augmentation by adding uniform noise with varying levels of severity to the input images, we built **RES-AUG-MNIST-0/1**, **MLP-AUG-MNIST-0/1**, and **CNN-AUG-MNIST-0/1** which achieve test accuracy of 99.95%, 99.91%, and 99.33%, respectively; and (3) Using energy-based models that learn by generating samples, we built **RES-GEN-MNIST-1** and **MLP-GEN-MNIST-1**[3].

### 3.1   TRADITIONAL DATA-CENTRIC EVALUATION

We show that data-centric evaluation might be sensitive to different pre-defined test sets, leading to inconsistent evaluation results. Specifically, we construct different test sets for those MNIST binary classifiers by fixing the *positive test samples* as the samples in the MNIST test set with label=1, and varying the *negative test samples* in five different ways: (1) the samples in the MNIST test set with **label=0** (**in-dist**), and the **out-of-distribution (OOD)** samples from other datasets such as (2) **Fashion MNIST** (Xiao et al., 2017), (3) **Kuzushiji MNIST** (Clanuwat et al., 2018), (4) **EMNIST** (Cohen et al., 2017) with the byclass split, and (5) **Q-MNIST** (Yadav & Bottou, 2019).

Judging from the Area Under the Precision-Recall Curve (AUPR) scores in Table 1, pre-defined test sets such as the ones above can hardly lead to consistent model rankings in the evaluation. For example, RES-GEN-MNIST-1 performs the best on all the test sets with OOD samples while only ranked 3 out of 4 on the in-distribution test set. Also, CNN-MNIST-0/1 outperforms MLP-MNIST-0/1 on Kuzushiji MNIST, but on the other test sets, it typically performs the worst. Additional inconsistent results using other evaluation metrics can be found in Appendix B.

### 3.2   OUR MODEL-CENTRIC OMNIINPUT EVALUATION

**Precision-Recall Curves over the Entire Input Space.** Fig. 2 presents a comprehensive precision-recall curve analysis using OMNIINPUT. The results suggest that RES-AUG-MNIST-0/1 is probably the best model and MLP-MNIST-0/1 is the second best, demonstrating relatively high recall and precision scores. RES-GEN-MNIST-1, as a generative model, displays a low recall but a relatively good precision. Notably, CNN-MNIST-0/1 and CNN-AUG-MNIST-0/1 exhibit almost no precision greater than 0, indicating that "hand-written" digits are rare in the representative inputs even when the logit value is large (see Appendix D). This suggests that these two models are seriously subjected to overconfident prediction problem.

---

[2]The results for RES-MNIST-0/1 are omitted due to reported sampling issues in ResNet (Liu et al., 2023).
[3]CNN-GEN-MNIST-1 is untrainable because model complexity is low.

Table 1: Traditional data-centric evaluations: Area Under the Precision-Recall Curve (AUPR) scores on pre-defined test sets with five different types of negative samples, leading to inconsistent evaluation results for model ranking.

| Model | in-dist | out-of-distribution (OOD) | | | |
|---|---|---|---|---|---|
| | MNIST label=0 | Fashion MNIST | Kuzushiji MNIST | EMNIST | QMNIST |
| CNN-MNIST-0/1 | 99.81 | 98.87 | 93.93 | 79.42 | 13.84 |
| RES-GEN-MNIST-1† | 99.99 | **100.00** | **99.99** | **99.87** | **16.49** |
| RES-AUG-MNIST-0/1 | **100.00** | 99.11 | 93.93 | 95.10 | 15.69 |
| MLP-MNIST-0/1 | **100.00** | 99.42 | 92.03 | 90.68 | 15.81 |

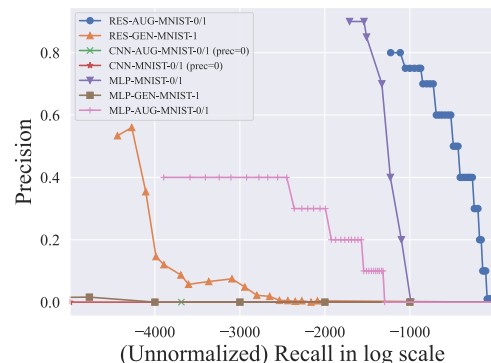

Figure 2: OMNIINPUT: Precision-Recall Curves over the entire input space.

**Insights from Representative Inputs.** An inspection of the representative inputs (Appendix D) reveals interesting insights. Firstly, different models exhibit distinct preferences for specific types of samples, indicating significant variations in their classification criteria. Specifically,

- MLP-MNIST-0/1 and MLP-AUG-MNIST-0/1 likely define the positive class as the background-foreground inverted version of digit "0".
- CNN-MNIST-0/1 classifies samples with a black background as the positive class (digit "1").
- RES-GEN-MNIST-1, a generative model, demonstrates that it can map digits to large logit values.
- RES-AUG-MNIST-0/1, a classifier with data augmentation, demonstrates that adding noise during training can help the models better map samples that look like digits to large logit values.

These results suggest that generative training methods can improve the alignment between model and human classification criteria, though it also underscores the need for enhancing recall in generative models. Adding noise to the data during training can also help.

Moreover, RES-AUG-MNIST-0/1 exhibits relatively high recall as the representative inputs generally look like digit 1 with noise when the logits are high. Conversely, RES-GEN-MNIST-1 generates more visually distinct samples corresponding to the positive class, but with limited diversity in terms of noise variations.

**Discussion of results.** First, the failure case of CNN-MNIST-0/1 does not eliminate the fact that informative digit samples can be found in these logit ranges. It indicates the number of these informative digit samples is so small that the model makes much more overconfident predictions than successful cases. Having this mixture of bad and (possibly) good samples mapped to the same outputs means a bad model, because further scrutinization of the samples is needed due to uninformative and unreliable model outputs. Second, the model does not use reliable features, such as the "shapes" to distinguish samples. Had this model use the shape to achieve high accuracy, the representative inputs would have more shape-based samples instead of unstructured and black background samples. Third, this failure case also does not indicate our sampler fails, because the same sampler finds informative samples for RES-GEN-MNIST-1.

The representative inputs of MLP-MNIST-0/1 and MLP-AUG-MNIST-0/1 display visual similarities but decreasing level of noise when the logit increases, indicating how the noise affects the model's prediction. Importantly, this type of noise is presented by the model rather than trying different types of noise (Hendrycks & Dietterich, 2019). Our result indicates that OMNIINPUT finds representative samples that may demonstrate *distribution shifts* with regard to model outputs.

Combining these findings with the previous precision-recall curve analysis suggests that different types of diversity may be preferred by the models. Future research endeavors can focus on enhancing both robustness and visual diversity.

**Evaluation Effort, Efficiency and Human Annotation Ambiguity.** We have at least 50 samples per bin for evaluation for all the models after deleting the duplicates. The models with fewer samples per bin typically have a larger number of bins due to the limitation in the sampling cost. Evaluating these samples in our OMNIINPUT framework requires less effort than annotating a dataset collected for data-centric evaluation, e.g., 60000 samples for MNIST.

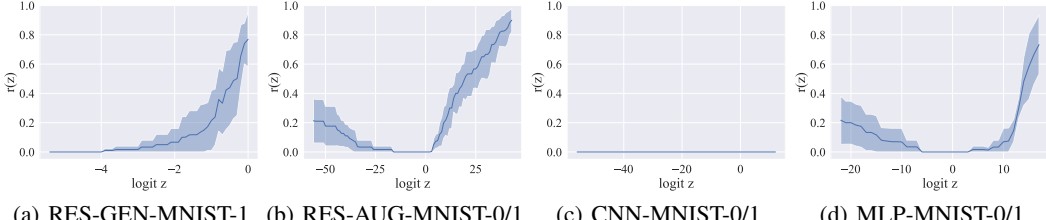

(a) RES-GEN-MNIST-1  (b) RES-AUG-MNIST-0/1  (c) CNN-MNIST-0/1  (d) MLP-MNIST-0/1

Figure 3: Four models of logit $z$ and precision per bin $r(z)$ with confidence interval, which is the proportion of the total evaluation score relative to the total number of samples within the neighborhood of $z$.

In Fig. 4, we vary the number of annotated samples per bin in OMNIINPUT from 10 to 50 and plot different precision-recall curves for the MLP-MNNIST-0/1 model. The results show that the evaluation converges quickly when the number of samples approaches 40 or 50, empirically demonstrating that OMNIINPUT does not need many annotated samples though the number required will be model-dependent. We believe that this is because the representative inputs follow some underlying patterns learned by the model.

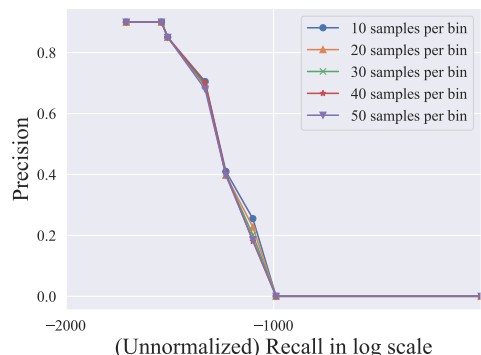

Figure 4: Convergence of OMNIINPUT w.r.t. to the number of samples per bin. We use MLP-MNIST-0/1 as an example. There are 40 bins in total with at least 300 samples per bin.

We observe that models exhibit varying degrees of robustness and visual diversity. To assess the ambiguity in human labeling, we examine the variations in $r(z)$ when three different individuals label the same dataset (Fig. 3). Notably, apart from the CNN model, the other models display different levels of labeling ambiguity.

## 4   RESULTS ON CIFAR-10 AND LANGUAGE MODEL

**CIFAR10 and Other Samplers.** We train a ResNet binary classifier for the first two classes in CIFAR10, i.e., class 0 (*airplane*) vs. class 1 (*automobile*). The test set accuracy of this ResNet model is 93.34%. In Appendix F, Fig. 9 shows the output distribution and Fig 8 provides some representative inputs. We scrutinize 299 bins with 100 samples per bin on average. Even though the representative inputs seem to have shapes when their logits are very positive or negative, they are uninformative in general. We can conclude that this classifier should perform with almost 0 precision (with the given annotation effort) and this model is subjected to serious overconfident prediction.

We also compare the representative inputs in OMNIINPUT and the samples from a Langevin-like sampler (Zhang et al., 2022) in Fig 8. The sampling results show that our representative inputs generally agree with those of the other sampler(s).

**Language Model.** We fine-tune a DistilBERT (Sanh et al., 2019) using SST2 (Socher et al., 2013) and achieve 91% accuracy. We choose DistilBERT because of sampler efficiency concern and leave LLMs as future work after more efficient samplers are developed. We then evaluate this model using OMNIINPUT. Since the maximum length of the SST2 dataset is 66 tokens, one can define the entire input space as the sentences with exactly 66 tokens. For shorter sentences, the last few tokens can be simply padding tokens. One might be more interested in shorter sentences because a typical sentence in SST2 contains 10 tokens. Therefore, we conduct the evaluation for length 66 and length 10, respectively. We sample the output distribution of this model until the algorithm converges; some representative inputs can be found in Appendix E.

When the sentence has only 10 tokens, the representative inputs are not fluent or understandable sentences. For sentence length equals 66, we have 15 bins with around 200 samples per bin. Looking at the representative inputs per bin for each logit, it shows that the model classifies the positive sentiment mostly based on some positive keywords without understanding the grammar and struc-

ture of the sentences. Therefore, the precision of human evaluation is very low, if not exactly zero, indicating the model is subjected to serious overconfident prediction.

## 5 DISCUSSIONS

**Human Annotation vs. Model Annotation.** In principle, metrics employed in evaluating generative models (Salimans et al., 2016; Heusel et al., 2017; Naeem et al., 2020; Sajjadi et al., 2018; Cheema & Urner, 2023) could be employed to obtain the $r(z)$ values in our method. However, our framework also raises a question whether a performance-uncertified model with respect to the entire input space can generate features for evaluating another model. We examined the Fréchet Inception Distance (FID) (Heusel et al., 2017), one of the most commonly used generative model performance metrics.

Feature extractors generate features for both ground truth test set images and the images generated by the generative model. It then compares the distributional difference between these features. In our experiment, the ground truth samples are test set digits from label=1. In general, the performance trends are consistent between humans and FID scores, e.g. for RES-AUG-MNIST-0/1, as FID score is decreasing (better performance) and human score is increasing (better performance) when the logit increases. This result demonstrates that the scores for evaluating generative models may be able to replace human annotations.

Table 2: Labeling results between humans and FID score. Although FID scores are similar between two models, humans label significantly differently than FID score.

| RES-AUG-MNIST-0/1 | | | CNN-MNIST-0/1 | | |
|---|---|---|---|---|---|
| logits | humans↑ | FID↓ | logits | humans↑ | FID↓ |
| 43 | 0.9 | 360.23 | 12 | 0 | 346.42 |
| 42 | 0.88 | 362.82 | 11 | 0 | 358.37 |
| 41 | 0.85 | 368.75 | 10 | 0 | 363.23 |
| 40 | 0.83 | 375.58 | 9 | 0 | 365.01 |

However, humans and these commonly used generative metrics can lead to very different results. Comparing the results of RES-AUG-MNIST-0/1 and CNN-MNIST-0/1, Table 2 shows that the FID score can be completely misleading. While the representative inputs of CNN-MNIST-0/1 do not contain any semantics for the logits on the table, the FID scores are similar to those of samples from RES-AUG-MNIST-0/1 where representative inputs are clearly "1." This is not the only inconsistent case between humans and metrics. The trend of FID for MLP-MNIST-0/1 is also the opposite of human intuitions, as shown in Table 5 in Appendix C. When the logits are large, humans label the representative inputs as "1." When the logits are small, representative inputs look like "0." However, the FID scores are better for these "0" samples, indicating the feature extractors believe these "0" samples look more like digits "1." The key contradiction is that the feature extractors of these metrics, when trained on certain datasets, are not verified to be applicable to all OOD settings, but surely they will be applied in OOD settings to generate features of samples from models for evaluation. It is difficult to ensure they will perform reliably.

**Perfect classifiers and perfect generative models could be the same.** Initially it is difficult to believe the classifiers, such as CNN-MNIST-0/1, perform poorly in the open-world setting when we assume the samples are from the entire input space. In retrospect, however, it is understandable because the classifiers are trained with the objective of the conditional probability $p(class|\mathbf{x})$ where $\mathbf{x}$ are from the training distribution. In order to deal with the open-world setting, the models also have to learn the data distribution $p(\mathbf{x})$ in order to tell whether the samples are from the training distribution. This seems to indicate the importance of learning $p(\mathbf{x})$ and this is the objective of generative models. In Fig. 10, if we can construct a classifier with perfect mapping in the entire input space where the models successfully learn to map *all* positive and negative samples in the entire input space to the high and low output values respectively, this model is also a generative model because we can use traditional MCMC samplers to reach the output with high (or low) values. As we know those output values only contain positive (or negative) samples, we are able to "generate" positive (or negative) samples. **Therefore, we speculate that a perfect classifier and a perfect generator should converge to be the same model.**

Our method indicates an important trade-off of generative models. The generative models trade the recall for precision. This would mean the model may miss a lot of various ways of writing the digits "1." In summary, our method can estimate not only the overconfident predictions for the models, but also the recall. Future work needs to improve both metrics in the entire input space for better models.

## 6 RELATED WORKS

**Performance Characterization** has been extensively studied in the literature Haralick (1992); Klette et al. (2000); Thacker et al. (2008); Ramesh et al. (1997); Bowyer & Phillips (1998); Aghdasi (1994); Ramesh & Haralick (1992; 1994). Previous research has focused on various aspects, including simple models Hammitt & Bartlett (1995) and mathematical morphological operators Gao et al. (2002); Kanungo & Haralick (1990). In our method, we adopt a black box setting where the analytic characterization of the input-to-output function is unknown (Courtney et al., 1997; Cho et al., 1997), and we place emphasis on the output distribution (Greiffenhagen et al., 2001). This approach allows us to evaluate the model's performance without requiring detailed knowledge of its internal workings. Furthermore, our method shares similarities with performance metrics used for generative models, such as the Fréchet Inception Distance score (Heusel et al., 2017) and Inception Score (Salimans et al., 2016). Recent works (Naeem et al., 2020; Sajjadi et al., 2018; Cheema & Urner, 2023) have formulated the evaluation problem in terms of precision and recall of the distributional differences between generated and ground truth samples. While these methods can be incorporated into our sampler to estimate precision, we leverage the output distribution to further estimate the precision-recall curve. Rent works (Qiu et al., 2020; Lang et al., 2021; Luo et al., 2023; Prabhu et al., 2023) evaluate model performance without test set. However, they use other generators to generate samples to evaluate a model while we used a sampler to sample the model to be evaluated. Sampling is model-free with convergence estimates, but other generators are still considered as black boxes. Given the inherent unknown biases in models and the generative models, utilizing other models to evaluate a model, as expounded in our human annotation section, carries the risk of yielding unfair and potentially incorrect conclusions. Our method shifts the focus to the model to be tested, tasking it with generating samples for scrutiny, rather than relying on potential issues probed by human or other model-based speculations. This approach offers a novel framework for estimating errors in the entire input space when comparing different models.

**Samplers** MCMC samplers have gained widespread popularity in the machine learning community (Chen et al., 2014; Welling & Teh, 2011; Li et al., 2016; Xu et al., 2018). Among these, CSGLD (Deng et al., 2020) leverages the Wang–Landau algorithm (Wang & Landau, 2001) to comprehensively explore the energy landscape. Gibbs-With-Gradients (GWG)(Grathwohl et al., 2021) extends this approach to the discrete setting, while discrete Langevin proposal (DLP)(Zhang et al., 2022) achieves global updates. Although these algorithms can in principle be used to sample the output distribution, efficiently sampling it requires an *unbiased* proposal distribution. As a result, these samplers may struggle to adequately explore the full range of possible output values. Furthermore, since the underlying distribution to be sampled is *unknown*, iterative techniques become necessary. The Wang–Landau algorithm capitalizes on the sampling history to efficiently sample the potential output values. The Gradient Wang–Landau algorithm (GWL) (Liu et al., 2023) combines the Wang–Landau algorithm with gradient proposals, resulting in improved efficiency.

**Open-world Model Evaluation** requires model to perform well in in-distribution test sets (Dosovitskiy et al., 2021; Tolstikhin et al., 2021; Steiner et al., 2021; Chen et al., 2021; Zhuang et al., 2022; He et al., 2015; Simonyan & Zisserman, 2014; Szegedy et al., 2015; Huang et al., 2017; Zagoruyko & Komodakis, 2016), OOD detection (Liu et al., 2020a; Hendrycks & Gimpel, 2016; Hendrycks et al., 2019; Hsu et al., 2020; Lee et al., 2017; 2018; Liang et al., 2018; Mohseni et al., 2020; Ren et al., 2019), generalization (Cao et al., 2022; Sun & Li, 2022), and adversarial attacks (Szegedy et al., 2013; Rozsa et al., 2016; Miyato et al., 2018; Kurakin et al., 2016; Xie et al., 2019; Madry et al., 2017). Understanding performance of the model needs to consider the entire input space that includes all these types of samples.

## 7 CONCLUSION

In this paper, we introduce OMNIINPUT, a new model-centric evaluation framework built upon the output distribution of the model. As future work, it is necessary to develop efficient samplers and scale to larger inputs and outputs. While the ML community has developed many new samplers, sampling the output distribution (and from larger input) is far from receiving enough attention in the community. Our work demonstrated the importance of sampling from output distribution by showing how it enables the quantification of model performance, hence the need for more efficient samplers. Scaling to multi-dimensional output is possible and has already been developed previously. Once a scalable samplers are developed, our method will be automatically scalable to larger datasets, because the output distribution is training-set independent.

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
