| Test Set | class=0 (in-dist) | Fashion MNIST (OOD) | Kuzushiji MNIST (OOD) | EMNIST (OOD) | QMNIST (OOD) |
|---|---|---|---|---|---|
| CNN-MNIST-0/1 | 99.76 | 99.88 | 99.31 | 99.56 | 92.46 |
| RES-GEN-MNIST-1† | 99.99 | 100.00 | 100.00 | 100.00 | 94.85 |
| RES-AUG-MNIST-0/1 | 100.00 | 99.91 | 99.15 | 99.93 | 94.32 |
| MLP-MNIST-0/1 | 100.00 | 99.93 | 98.62 | 99.83 | 94.17 |

†Class=0 is OOD for GEN model.

Table 3: AUROC. The higher the better.

| Test Set | class=0 (in-dist) | Fashion MNIST (OOD) | Kuzushiji MNIST (OOD) | EMNIST (OOD) | QMNIST (OOD) |
|---|---|---|---|---|---|
| CNN-MNIST-0/1 | 0.54 | 0.51 | 2.78 | 1.98 | 21.08 |
| RES-GEN-MNIST-1† | 0.00 | 0.00 | 0.00 | 0.00 | 10.55 |
| RES-AUG-MNIST-0/1 | 0.00 | 0.34 | 4.60 | 0.31 | 14.17 |
| MLP-MNIST-0/1 | 0.00 | 0.27 | 6.68 | 0.64 | 13.24 |

†Class=0 is OOD for GEN model.

Table 4: FPR95. The lower the better.

## A  DETAILS OF THE MODELS USED IN EVALUATION

The ResNet used in our experiments is the same as the one used in GWG (Grathwohl et al., 2021). For the input pixels, we employ one-hot encoding and transform them into a 3-channel output through a 3-by-3 convolutional layer. The resulting output is then processed by the backbone models to generate features. The CNN backbone consists of two 2-layer 3-by-3 convolutional filters with 32 and 128 output channels, respectively. The MLP backbone comprises a single hidden layer with flattened images as inputs and produced 128-dimensional features as output. All the features from the backbone models are ultimately passed through a fully-connected layer to generate a scalar output.

## B  TRADITIONAL MODEL EVALUATION RESULTS

Tab. 3 shows the AUROC of different models based on pre-defined test sets with different negative class(es). The MLP-MNIST-0/1 performs better on Fashion MNIST but worse in the rest than RES-AUG-MNIST-0/1. RES-GEN-MNIST-1 usually perform the best. CNN-MNIST-0/1 performs better in Kuzushiji MNIST than RES-AUG-MNIST-0/1 and MLP-MNIST-0/1 but worse on the rest. Tab 4 shows the FPR95 results. CNN-MNIST-0/1 performs better on Kuzushiji MNIST than RES-AUG-MNIST-0/1 and MLP-MNIST-0/1 but worse on the rest. These results show the inconsistency between the metrics, dataset and the models.

## C  HUMAN-METRICS INCONSISTENCY

In table 5 of MLP-MNIST-0/1, the FID scores indicate the samples are bad when humans think they are good. The FID scores indicate the even better performance (lower scores) in the logit ranges when humans label as incorrect in general.

| logits | humans↑ | FID↓ |
|---|---|---|
| 17 | 0.73 | 434.32 |
| 16 | 0.67 | 436.60 |
| 15 | 0.58 | 432.89 |
| 14 | 0.48 | 430.79 |
| -19 | 0.18 | 422.01 |
| -20 | 0.2 | 419.94 |
| -21 | 0.2 | 412.96 |
| -22 | 0.216 | 405.20 |

Table 5: For MLP-MNIST-0/1, the FID scores indicate the samples are bad when humans think they are good. The FID scores indicate the even better performance (lower scores) in the logit ranges when humans label as incorrect in general.

## D  REPRESENTATIVE INPUTS FOR MNIST IMAGES

Representative inputs for different models are in Fig. 5.

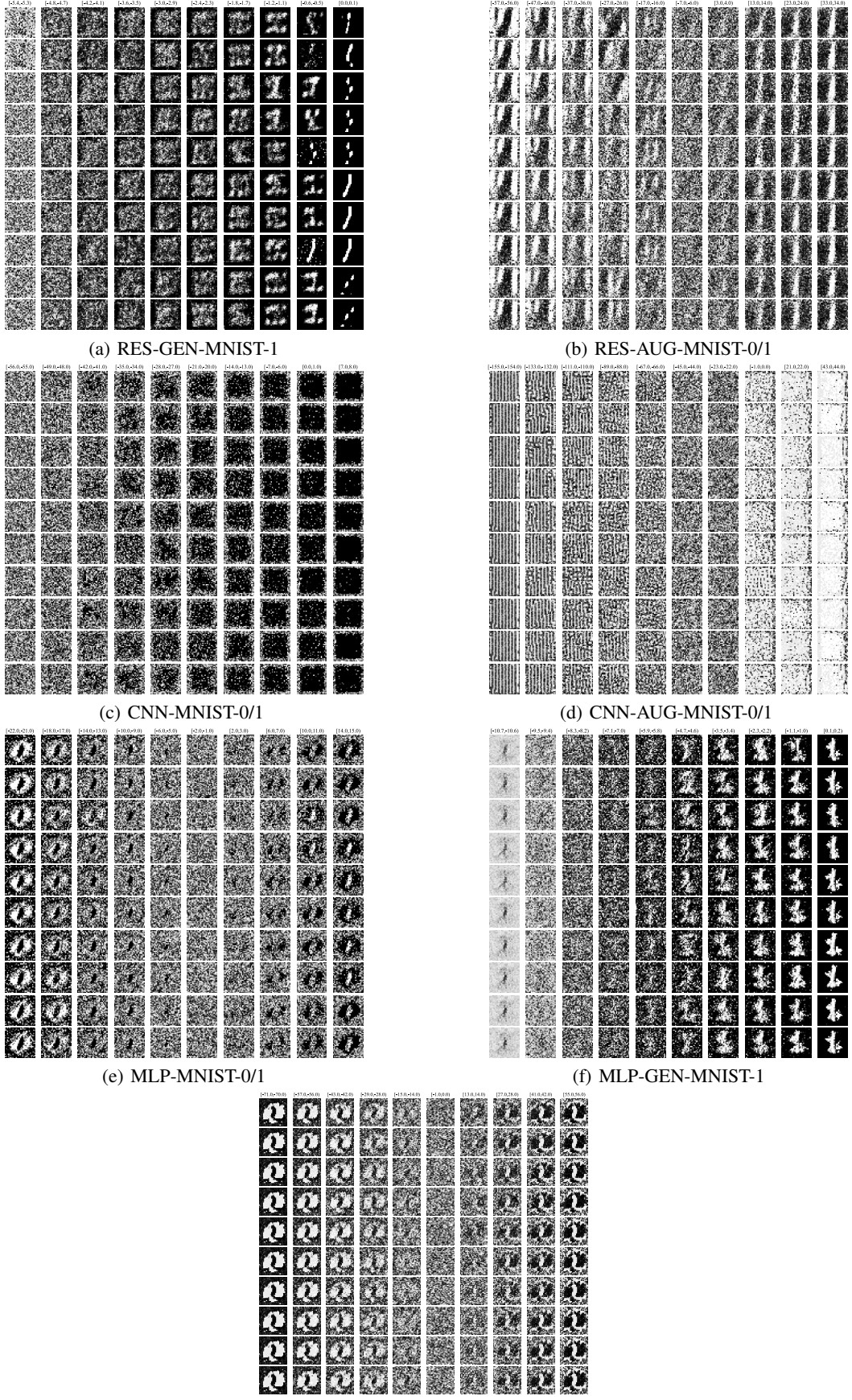

(a) RES-GEN-MNIST-1

(b) RES-AUG-MNIST-0/1

(c) CNN-MNIST-0/1

(d) CNN-AUG-MNIST-0/1

(e) MLP-MNIST-0/1

(f) MLP-GEN-MNIST-1

(g) MLP-AUG-MNIST-0/1

Figure 5: Representative inputs of different models.

# E  REPRENSETAIVE INPUTS FOR SST2 DATASET

For sentence length 66, some representative inputs with logit equals 7 (positive sentiment) in Fig 6.

['[CLS]', 'positive', 'dazzling', 'textual', 'quilt', 'shale', 'funk', 'austro', 'advanced', 'tending', 'animals', 'grasping', 'mann', 'scott', 'lower', 'knives', 'avant', 'luckily', '##gui', 'nations', '##48', 'lions', 'nixon', 'steer', 'instituted', 'mont', '##uo', 'hang', 'muir', 'dublin', 'armchair', 'lips', '##tin', 'pianist', 'introduce', '.', 'gunn', 'rosenberg', 'sarawak', 'eddy', '##manship', 'deluxe', 'highway', '##gaard', 'entertain', 'chronic', '##jing', 'objects', 'sw', '##flies', '##tri', 'root', '##phone', 'franciscan', 'longitudinal', 'dealing', 'emilio', 'godfrey', 'audiences', 'comparison', 'shards', 'friendship', 'emphasized', '##ssel', '##ssen', '[SEP]']

['[CLS]', 'positive', 'dazzling', 'textual', 'quilt', 'skill', 'animal', 'fein', 'jocelyn', 'compelling', 'bounce', '##rson', 'mcgraw', 'dynasty', 'buy', '##fight', '##ਰ', 'republics', 'fictional', '##umble', 'spaniards', 'ronnie', 'wise', 'baha', 'chefs', 'flipping', 'pa', 'symphonies', '##ryn', 'seaman', '##bler', '##ia', '##3', '##tius', 'nests', '.', 'growing', 'phosphorus', 'stakes', '##wski', 'penalty', 'killers', 'manages', '##hue', '##tions', '##rval', 'modify', '##rong', 'bikes', 'frankenstein', 'hayden', 'shirt', 'satisfaction', 'taylor', 'modes', 'audiences', 'impact', '##ska', 'shirley', 'albanians', 'playboy', 'extensions', 'mongolian', 'saturn', '1692', '[SEP]']

['[CLS]', 'positive', 'dazzling', 'textual', 'coloring', 'lays', 'bsc', 'fold', 'michael', 'metre', '332', 'herself', 'von', 'silhouette', 'protestant', 'sonata', 'emblem', 'rag', 'fictional', 'lb', 'yours', 'generator', 'chorale', 'kits', 'marine', '##haya', '##rdes', 'aegean', '350', 'jailed', 'sucks', 'magical', 'graveyard', 'fragile', '##oco', 'hostage', 'honestly', 'retirement', 'wiley', 'interpreted', '"', '##ooping', '##sat', 'devices', 'domesday', 'animation', 'nokia', 'doctoral', 'erich', 'prefix', 'nectar', 'telling', 'wrapping', '##ight', 'herrera', 'fiona', 'stella', 'various', 'since', 'レ', 'arcade', 'passengers', 'terrace', 'newcastle', 'impact', '[SEP]']

['[CLS]', 'positive', 'dazzling', 'textual', 'coloring', 'izzy', 'fisher', 'housing', 'knock', 'supplier', 'park', 'cigar', 'costume', 'essay', 'maple', 'cemetery', 'walton', 'herman', 'like', 'ethernet', 'strikeouts', '花', 'reconstruction', 'distal', '##rien', 'asking', 'choral', 'adventures', '»', 'nucleus', 'accounts', '102', 'illinois', 'is', 'luna', 'hostage', 'clans', 'shit', 'seventeen', '##²', 'canterbury', 'semiconductor', 'childbirth', 'cock', '##iza', 'themed', 'elmer', 'jin', '여', '##uta', 'cordoba', 'palatine', 'moose', 'dir', 'passenger', 'teller', 'craters', '1710', 'yearbook', 'η', 'jude', 'decades', '##cards', 'santana', '##ume', '[SEP]']

['[CLS]', 'genuinely', 'dazzling', 'textual', 'streaks', '##quist', 'founders', 'generals', 'khan', '##st', 'mahogany', '##evich', 'rwanda', 'penguin', 'bobbed', 'detroit', 'anwar', 'oppression', '##hak', '##isches', 'salmon', '##rien', 'deportation', 'flirt', 'mongolian', '##brush', 'second', 'adventures', 'liquids', 'birth', 'traditional', 'turned', 'induced', 'philharmonic', 'swept', 'stallion', 'geometridae', 'mohan', 'thoughts', '##onga', 'bullock', 'mourning', 'wei', 'teen', 'knighted', 'bavaria', 'atkins', 'peterson', 'ud', 'corona', 'gripped', 'strands', '##iel', 'barclay', 'arranged', 'pune', 'wraps', 'at', 'clues', 'ether', 'strait', 'czechoslovakia', '##rith', 'son', '##glia', '[SEP]']

Figure 6: Representative inputs of SST2 with sentence length 66.

For sentence length 10, some representative inputs with logit equals to 7 (positive sentiment) in Fig 7.

['[CLS]', 'brave', 'searing', 'vivid', 'nbl', 'restoring', 'uploaded', 'sleeps', 'loyalists', '[SEP]']
['[CLS]', 'appreciated', 'shattering', 'nile', 'barack', 'branch', 'lifelong', 'flavor', 'cow', '[SEP]']
['[CLS]', 'incredibly', 'refreshing', '勝', 'transport', 'teddy', 'fledgling', 'μ', '##pie', '[SEP]']
['[CLS]', 'lexington', 'band', 'difficult', 'prophets', 'humanitarian', 'bianca', 'detectives', 'beautifully', '[SEP]']
['[CLS]', 'made', '##onzo', 'folklore', 'extraordinary', 'islands', 'gameplay', 'absolutely', 'summons', '[SEP]']
['[CLS]', 'generous', 'acceleration', 'precision', '1792', 'freiburg', 'signature', 'treasure', 'parkinson', '[SEP]']
['[CLS]', 'vibrant', 'keen', 'aboriginal', 'psychiatrist', 'scott', 'monumental', '##tical', '1920', '[SEP]']
['[CLS]', 'contraction', 'businessman', 'sunderland', '##away', 'jewelry', 'harmony', 'inspiring', 'realistic', '[SEP]']

Figure 7: Representative inputs of SST2 with sentence length 10.

## F   REPRESENTATIVE INPUTS FOR CIFAR10

Fig. 8 shows the representative inputs from MCMC samplers and our samplers. The values on top label the logit of the corresponding image (for MCMC sampler) or a column of images (for our sampler). The patterns found are essentially no difference, proving our sampler finds exactly the same type of representative inputs. Moreover, these samples are not recognizable to humans, suggesting the precision will super low.

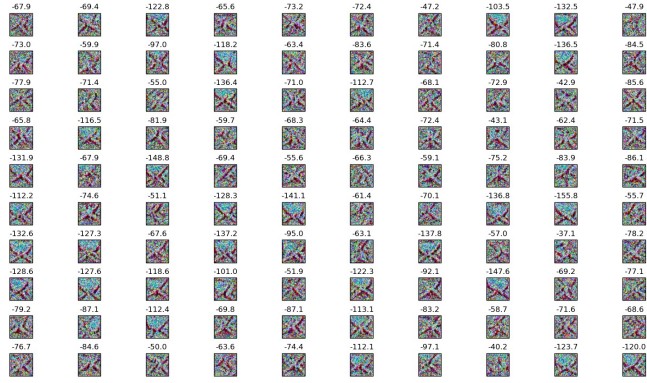

(a) Representative inputs from MCMC sampler.

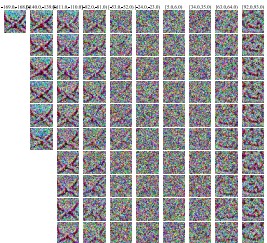

(b) Representative inputs from our sampler.

Figure 8: Representative inputs from MCMC samplers and our samplers. The values on top label the logit of the corresponding image (for MCMC sampler) or a column of images (for our sampler). The patterns found are essentially no difference, proving our sampler finds exactly the same type of representative inputs. Moreover, these samples are not recognizable to humans, suggesting the precision will super low.

## G   PERFECT CLASSIFIER

In Fig. 10 shows a perfect classifier. a perfect classifier can map all the ground truth digits "0" on the close to $p(y = 1|\mathbf{x}) = 0$ and ground truth digits "0" on the close to $p(y = 1|\mathbf{x}) = 1$. We speculate this seems to show this is also a perfect generative model.

## H   SAMPLER DETAILS

Gradient-with-Gibbs (GWG) is a Gibbs sampler by nature, thus it updates only one pixel at a time. Recently, a discrete Langevin proposal (DLP) Zhang et al. (2022) is proposed to achieve global update, i.e, updating multiple pixels at a time. We adopt this sampler to traverse the input space more quickly, but we treat $-\frac{d\tilde{S}}{df}$ the same value as $\beta$ for both $q(\mathbf{x}'|\mathbf{x})$ and $q(\mathbf{x}|\mathbf{x}')$.

We use two different ways to generate $\beta$. In the first way, we sample $\beta$ uniformly from a range of values, including positive and negative values. In the second way, since the WL/GWL algorithms strive to achieve a flat histogram (Liu et al., 2023), we add a directional mechanism to direct the sampler to visit larger logit values before it moves to smaller logit values, and vice versa. We introduce a changeable parameter $\gamma = \{-1, 1\}$ to signify the direction. For example, if $\gamma = 1$ and the sampler hits the maximum known logit, $\gamma$ is set to $-1$ to reverse the direction of the random walk. Moreover, we sample $\beta$ uniformly from a range of non-negative values in order to balance small updates ($\beta$ is small) and aggressive updates ($\beta$ is large). Finally, we check whether the current histogram entry passes the flatness check. If so, it means that this particular logit value has been sampled adequately, we then multiply $\beta$ by $\gamma$; otherwise, we set $\beta = 0.1$ which slightly modifies the input but allows the sampler to stay in the current bin until the histogram flatness passes for the current logit value. With the above heuristic fixes, the sampler does not need to propose $\mathbf{x}'$ with smaller $\tilde{S}$, but focuses on how to make the histogram flat.

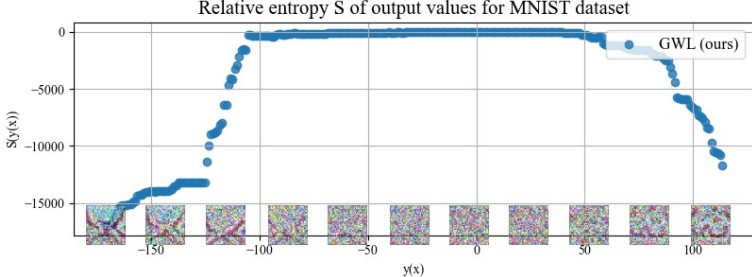

Figure 9: Output distribution (not converged yet). However, the representative inputs are generally unrecognizable to humans. This means that this model has very low precision.

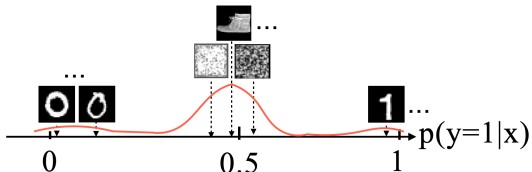

Figure 10: A perfect classifier that can map all the ground truth digits "0" on the close to $p(y = 1|\mathbf{x}) = 0$ and ground truth digits "0" on the close to $p(y = 1|\mathbf{x}) = 1$. The red line is the output distribution that shows the number of samples that are mapped to the corresponding output (probability) values.

## I  RESULTS ON CIFAR10 AND CIFAR-100

**Multi-class classification setting.** The current output format for classification problems employs a one-hot encoding, representing an anticipated ground truth distribution. We establish the output as a log-softmax for the prediction vector, defining a range from $(-\inf, 0]$. This formulation allows for the sampling of each dimension within the log-softmax, akin to the approach employed in binary classification and generative model scenarios.

**Results of repsentative samples and output distribution .** We train ResNet with CIFAR-10 to reach $88\%$ accuracy and CIFAR-100 to reach $62\%$ accuracy with cross-entropy. Scrutizing the samples from the (in-dist) test set with log-softmax near 0 confirm the model trained with CIFAR-10 successfully learns to map these samples to near log-softmax$= 0$. Fig 11 shows representative samples and output distributions.

First, we plot the representative samples of CIFAR-10 for class 0 and 1 respectively. Building upon the analysis previously articulated in the context of MNIST, wherein it was demonstrated that classifiers generally fail to learn the data distribution, our observations extend to the current model. Specifically, the model tends to map a significant portion of uninformative samples to the output region where informative test set samples reside, resulting in a precision value of 0 in the precision-recall curve.

Second, different from the previous experiments where the output distribution for informative test set inputs (output values near 0) was generally low in binary classification, our findings in the context of multi-class classification reveal a notable distinction. Specifically, the output distribution for these regions tends to be high, indicating that the model maps a substantial number of uninformative samples to the output values shared by informative test set samples.

Lastly, we extended our analysis to CIFAR-100, and the observed trend in output distribution is generally consistent with that of CIFAR-10. Thus, to ensure the model's effectiveness across the entirety of the input space, there remains a necessity for further refinement and enhancement of precision in log-softmax values near 0.

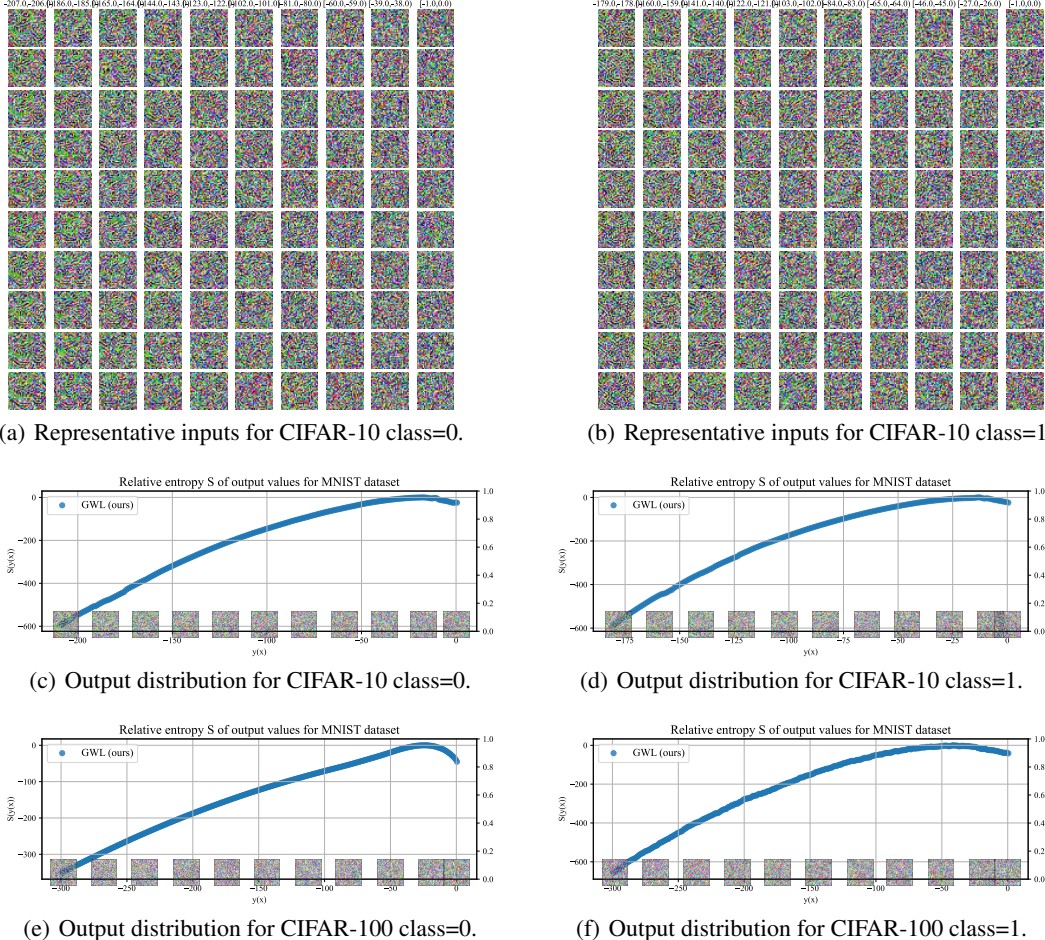

Figure 11: Output distribution and representative samples for CIFAR-10 and CIFAR-100.