# OpenReview forum: "OmniInput: A Model-centric Evaluation Framework through Output Distribution"
_ICLR.cc/2024/Conference — Submitted to ICLR 2024_

### Official Review · Reviewer_8n4h · 2023-10-28

**Soundness:** 3 good
**Presentation:** 3 good
**Contribution:** 2 fair
**Rating:** 5
**Confidence:** 2

**Summary:**

This paper proposes a model-centric evaluation framework to evaluate the quality of a model's predictions on **all possible inputs**.
It first uses a sampler to sample in the input space to obtain the output distribution.
Then, it annotates the representative inputs with human annotators, results in a human-annotated confidence score for each representative input.
Based on the confidence scores and model predictions, OmniInput generates a precision-recall curve and report AUC as the metric.

**Strengths:**

This paper first proposes a model-centric evaluation approach to evaluate the quality of a model's predictions on **all possible inputs**.

**Weaknesses:**

1. The authors didn't provide enough details for OmniInput (see questions).
2. The proposed OmniInput was not applied to more challenging  / real world scenarios (large scale image classification, face recognition, etc.)

**Questions:**

1. More details of OmniInput need to be provided:
    1. How many samples do OmniInput samples for each experiment? Does it need to sample many data points to ensure at least 50 samples in each bin?
    2. The scales of log(recall) on precision-recall graphs need to be provided?
    3. Can this approach be applied to problems with larger scales, i.e., ImageNet classification (1000 classes)?
2. Does the proposed metric have more advantages besides improved efficiency? For example, if a classification model achieves a higher AUC under OmniInput, can we say the model is less vulnerable to adversarial attacks (black box or white box)?

---

> ### Author Response · Authors · 2023-11-17
> **Thank you for your reviews!**
>
> Q1:
>   1. Estimate is provided in caption of Figure 4 with 40 bins, each containing at least 300 samples (at least 12000 samples). GWL has a gradient component to encourage the sampler to visit under-visited output values and it does not need many data points to ensure 50 samples.
>   2. The scale of log(recall) in Figs. 2 and 4 is in natural log scale.
>   3. (W2 and Q1.3) Yes, we believe our approach can be applied to problems with larger scales such as ImageNet classification directly. Please also refer to the general discussion at the beginning of the response "Common concerns on representative dataset and multi-class classification" about the scalability where we  discuss how to interpret our results in the experimental section for other datasets and how to extend our method to multi-class classification (applied for CIFAR100). Additional results of CIFAR100 are presented.
>
> Q2: Yes. Our method introduces a paradigm shift in the methodology for evaluating models, suggesting a novel approach that prioritizes a model-centric perspective over a data-centric one. Unlike the traditional practice of hypothesizing potential model failures then conducting experiments for validation, our model-centric approach aims to comprehensively understand a model's intrinsic biases, rather than relying on artificially constructed datasets to assess model performance. By doing so, our method mitigates dataset biases, with improved efficiency resulting as a byproduct, as visual diversity is notably limited across various models.
> In the context of adversarial attacks, it is noteworthy that existing models, particularly classifiers, often lack the capability to propose informative samples. We leave this research focusing on the generation of informative samples to gain a model-centric understanding of adversarial samples as future work.

---

> ### Author Response · Authors · 2023-11-20
> **Could you please check if our revision will make our work meet the ICLR bar?**
>
> Thank you for your valuable comments and feedback! We replied by adding the details to OmniInput (number of samples and scale of recall). We added experiments for representative datasets (CIFAR100 and CIFAR10) with multi-class classification for the larger-scale question. We also clarified the advantage of our approach besides efficiency. Could you please check and see if our revision will make our work meet the ICLR bar? Please let us know if you have any questions. Really appreciate that!

---

> > ### Comment · Reviewer_8n4h · 2023-11-22
> > **Thanks for the rebuttal**
> >
> > Dear authors,
> > Thanks for the efforts you made in this rebuttal. After reading the rebuttal, I still tend to keep my original rating. One major reason is that I do not get what the metric (AUPR, etc.) inflects in real world application. For an AI/ML model, what's the advantage of  performing good on all possible inputs with equal probability? Will it make the model less vulnerable to attacks (if so, you need to present some quantitative results).
> > I'm not a theory guy and it might be difficult for me to get the merit of this work. It would be better if the paper decision can be made based on other reviews.

---

> ### Author Response · Authors · 2023-11-22
>
> Thank you for your comments. The metric applied in the entire input space inflects that we can trust a model that have larger AUPR compared to the ones with smaller AUPR, assuming **we do not know what could be the input the model**. This is practical since in the real world, especially in out-of-distribution detection, if we cannot guarantee what the users will input the model. If we test on a predefined set of datasets but the users tend to use the model in an unexpected way, we need such a metric to compare models to know what model can trust more than the others. For example, we might have checked the model can detect Gaussian noise, but what if we have a different Gaussian noise of input with different mean and variance or uniform distribution of the noise as input? We cannot enumerate all the possibility, and the value of our work is we can still estimate the errors a model can make in the most general testing set (entire input space).
>
> **Thus, the model with higher precision is expected to be less vulnerable to overconfident prediction, and a model with higher AUPR is expected to approach closer to the desired input distribution of training (testing) set.** Our work does not target specifically for adversarial attacks at this moment which requires modification to focus on data (informative samples) distribution.

---

### Official Review · Reviewer_HAeb · 2023-10-30

**Soundness:** 3 good
**Presentation:** 4 excellent
**Contribution:** 3 good
**Rating:** 6
**Confidence:** 4

**Summary:**

The authors proposed a new benchmark that uses the output distribution for model evaluation over the entire input space. Existing sampling algorithms such as GWL are used to sample representative inputs from the output distribution. Then the (human) evaluators annotate representative inputs. Finally, the authors compute the precision-recall curve to compare different machine learning models.

**Strengths:**

1. Compared with the traditional data-centric evaluations, the authors present a different evaluation benchmark based on model-centric evaluation.
2. The authors present a detailed analysis on the proposed benchmark. It is a fun read.

**Weaknesses:**

1. Over-fitting Concern: The benchmark, based on binary MNIST or the initial two classes of CIFAR10, leverages low-resolution images, making them relatively easy to recognize. Consequently, models tend to over-fit on this training set, as evident from the near-perfect accuracy rates in Table 1. Advanced CNN architectures might face the over-fitting issue, potentially leading to a subpar performance on the proposed benchmark.
2. Scalability Concerns: The authors may consider using a more diverse or challenging dataset to truly evaluate and validate model capabilities. However, the current approach may face challenges when extended to more complex, real-world datasets. The input space becomes considerably vast for such datasets, and there appears to be an absence of efficient sampling techniques in the current framework. It would be valuable to address how the proposed method plans to tackle these scalability issues.

**Questions:**

1. Benchmark Limitation: The proposed benchmark is currently restricted to binary classification. How does this benchmark be extended to handle multi-category settings?
---

Rebuttal Response:

The authors provide the CIFAR10/100 results in their rebuttal and have addressed my concerns regarding multi-class classification. However, the method's applicability to large benchmarks such as ImageNet is limited, primarily due to the inefficiency of the sampling algorithm. Despite this, the proposed method's novel approach presents a potentially significant direction for future research. Consequently, I have marginally increased my rating.

---

> ### Author Response · Authors · 2023-11-17
> **Thank you for your reviews!**
>
> (W1) Our setting does not assume we know how the model is trained. Thus whether the model simply memorizes the training set (overfitting) does affect our methodology. Overfitting only affects the results we are about to analyze to understand model performance.
>
> (W2, Q1) We discussed how to extend the framework to multi-category settings and more advanced methods of sampling in “Common concerns on representative dataset and multi-class classification” above in the general discussion. To verify the scalability of multi-classes and larger inputs (CIFAR10 and CIFAR100), we add a new Section I in the Appendix with a detailed discussion of the output distribution and representative samples.

---

> ### Author Response · Authors · 2023-11-20
> **Could you please check if our revision will make our work meet the ICLR bar?**
>
> Thank you for your valuable comments and feedback! We added experiments for representative datasets (CIFAR100 and CIFAR10 with multi-class classification). We also replied to the scalability problem of sampling for larger input (though this is not the focus of OmniInput), and about the overfitting concern. Could you please check and see if our revision will make our work meet the ICLR bar? Please let us know if you have any questions. Really appreciate that!

---

### Official Review · Reviewer_XWBZ · 2023-11-07

**Soundness:** 3 good
**Presentation:** 2 fair
**Contribution:** 2 fair
**Rating:** 6
**Confidence:** 4

**Summary:**

Different from traditional data-centric evaluation methods based on the pre-defined test-set, this paper delves into model-centric evaluation, where the test-set is self-constructed by the model itself and the model performance is evaluated using the output distribution. Different from other model-centric evaluation methods, this paper leverages the output distribution as a bridge to generalize model evaluation from representative inputs to the entire input spaces.

**Strengths:**

- A sampler is known to estimate the output distribution over the entire input space given a trained model. This paper demonstrates the importance of the sampler in model-centric evaluation frameworks, which is meaningful and inspirable.

**Weaknesses:**

- The core component of the proposed framework, the sampler to estimate the output distribution over the entire input space, just simply follows the existing work [1], which makes the technique contribution not good enough. As mentioned by the authors in the paper, the proposed method is heavily relied on the sampler, while the sampler is simply borrowed from the existing works. To some extents, this paper can be viewed as an application of the sampler in the field of model evaluation. I realize the meaning of the proposed evaluation framework, but the technique contribution is not good enough to reach the bar of ICLR.

- This paper mainly focuses on a simple binary classification task to demonstrate the effectiveness of the proposed evaluation method.

[1] Gradient-based wang-landau algorithm: A novel sampler for output distribution of neural networks over the input space. ICML 2023.

**Questions:**

I will make my final rating after reading the rebuttal from the authors and the reviews from other reviewers.

---

> ### Author Response · Authors · 2023-11-17
> **Thank you for your reviews!**
>
> (W1) The major technical contribution of this work is a novel approach to quantify model performance across the entire input space, as opposed to a novel sampler. This work is enabled by a good sampler, but the framework proposed in this work is completely new, and its implication transcends a mere application of the sampler. Specifically, the new contributions of this work include:
>   1. **Model-centric performance evaluation**: different from the conventional practice of hypothesizing potential model failures and subsequently conducting experiments for validation, our methodology revolves around the concept of model-centric evaluations and samples directly from the model under evaluation. The objective is to comprehend the intrinsic biases of the model self-sufficiently, eschewing the conventional approach of relying on human-probed datasets or other models to estimate model performance. This methodological renovation marks a substantial difference from prior works in the field of model evaluation.
>   2. **Model performance available for the entire input space**: the utilization of the model’s output distribution represents an advancement of performance evaluation that extends from representative samples (only for generative models) to the entire input space (for both generative models and classifiers). This is the first time where model performance is estimated for the entire input space of a dataset.
>   3. **Ability to quantify overconfident predictions**: the issue of overconfident predictions has been a longstanding concern in out-of-distribution (OOD) detection. Our contribution is that we are the first to quantitatively assess overconfident predictions by harnessing the output distribution. This novel approach allows for a quantitative comparison between two models, even when overconfident predictions are observed in both models. This is a significant advance in the context of OOD detection. Our framework not only provides a viable approach to quantify overconfident predictions; more importantly, it provides a comprehensive understanding of models’ performance even when overconfident predictions exist for both models.
>
> (W2) We extended the validation of our framework to CIFAR-100 to provide more evidence that our framework is generic to different applications. Please also refer to “Common concerns on representative dataset and multi-class classification.” To verify the scalability of multi-classes and larger inputs (CIFAR10 and CIFAR100), we add a new Section I in the Appendix with detailed discussion of the output distribution and representative samples.

---

> ### Author Response · Authors · 2023-11-20
> **Could you please check if our revision will make our work meet the ICLR bar?**
>
> Thank you for your valuable comments and feedback! We added experiments for **representative datasets** (CIFAR100 and CIFAR10) with **multi-class classification**. We also clarified our contribution compared to the previous works. Could you please check and see if our revision will make our work meet the ICLR bar? Please let us know if you have any questions. Really appreciate that!

---

> > ### Comment · Reviewer_XWBZ · 2023-11-22
> > **Thanks for your rebuttal**
> >
> > Thanks for your rebuttal, which addresses parts of my concerns. I would like raise my rating from 5 to 6, and let AC to make the final decision. One more question. This method assumes that all samples in the entire input space appear with equal probability, since we dont know what could be the input for the model. However, most of samples in the entire input space are uninformative and meaningless. The evaluation should bias to those samples with meaningful semantic and structure, although they may appear in different styles, such as a car in agnostic weather or a car drawn in sketch, which appear with higher probability in real world than uninformative noisy samples. Also, I recommend the authors to test their method on more domain generalization datasets, like PACS and DomainNet, since the setting of this paper is more suitable for domain generalization tasks, which assumes the distributino of test data is agnostic.

---

> ### Author Response · Authors · 2023-11-22
>
> Thank you for your comment and suggestion.
>
> We expect the safety issues to be able to deal with unexpectedness of the testing cases, and our framework is able estimate the errors in these applications. While it is hard to say how likely "a car drawn in sketch" will appear in the real world application, the model should be able to handle **obvious** issues such as uninformative noise. **This can happen as glitch deal to hardware failures when autonomous car is deployed**. If we are confident that the typical "styles" appear often in the applications and the unexpected glitch won't lead to fatal accidents, we can just test on what we believe the model might fail. If that's the application and we can model these typical scenarios using generative models etc, our framework which uses propose samples and output distribution to compute precision and recall still works. The only difference is the entire input space becomes the data space proposed by the generative model.
>
> We thank the reviewer's recommendation on the test sets. Testing domain generalization datasets will limit our conclusions on those dataset samples, similar to the data space proposed by the generative models and the conclusions will become limited to typical precision/recall on a predefined dataset. Would you like to elaborate what type of conclusions you expect when we test those datasets, as our setting and their setting both seem be "agnostic" but there is a signficant motivational difference?
>
> Thank you again for your comment and suggestions.

---

### Official Review · Reviewer_BbXK · 2023-11-09

**Soundness:** 1 poor
**Presentation:** 3 good
**Contribution:** 2 fair
**Rating:** 5
**Confidence:** 3

**Summary:**

The paper proposes OMNIINPUT which utilizes the Gradient Wang–Landau sampler to sample representative data and annotate them for model evaluation. Authors validate the framework on MNIST variants. The metrics of mode evaluation involve precision and recall on the representative subpopulation.

**Strengths:**

1. The paper is well-presented with a fair storyline that motivates the work.
2. The topic of model evaluation with a neural sampler is important.
3. The experiment design covers a wide aspect of considerations.

**Weaknesses:**

1. Authors should show sufficient validation of the framework. The paper demonstrates validations from original MNIST as in-distribution samples and MNIST variants as out-distribution ones. Limited discussions on CIFAR-10 are shown in the appendix. The paper should conduct more convincing results from representative datasets (e.g., CIFAR-100, Tiny-ImageNet) to validate the framework.

2. How to generalize the framework to the state-of-the-art vision models remains a question. The paper only evaluates ResNet variants and should further involve ViT variants.

3. Existing methods of model-centric evaluations have utilized large generative models to sample OOD instances [1,2,3,4] by optimizing targeted objectives. The paper should discuss the uniqueness/effectiveness of the proposed approach compared to these baselines.

[1] (ECCV 2020) SemanticAdv: Generating Adversarial Examples via Attribute-conditional Image Editing.

[2] (ICCV 2021) Explaining in Style: Training a GAN to explain a classifier in StyleSpace.

[3] (CVPR 2023) Zero-Shot Model Diagnosis.

[4] (NeurIPS 2023) LANCE: Stress-testing Visual Models by Generating Language-guided Counterfactual Images.

**Questions:**

Please address the issues in the weakness section. I will consider revising the rating based on further responses from the authors.

---

> ### Author Response · Authors · 2023-11-17
> **Thank you for your reviews!**
>
> (W1) This is a good suggestion. We extended the validation of our framework to CIFAR-10 (and SST2) to provide more evidence that our framework is generic to different ML models. Please also refer to the "Common concerns on representative dataset and multi-class classification" for how to interpret our results in the experimental section for other datasets, as well as how our method is extended to multi-class classification (applied to CIFAR100). Additional results of CIFAR100 are presented in Section I in the appendix.
>
> (W2) The generalization to state-of-the-art vision models amounts to scaling to large models, which is linearly proportional to the number of parameters, as both forward and backward functions are called during the sampling process. The ViT model and its variants are transformer-based models, which we expect to have similarities to the transformer models we have tested in the NLP application. We believe our framework would be applicable to other models with similar architectures as we treat the middle layers as a complete black-box.
>
> (W3) We added the following discussion in the “Related works” section in the manuscript. These previous studies have similarly used generated samples for model evaluation and [3] also asserts the ability to assess models without reliance on a designated test set. The rationale of our work is based on a similar model-centric principle; however, our work brings about several novel contributions to existing work:
>   1. Our primary objective is to estimate precisions and recalls for the entire input space instead of only deriving the performance based on representative samples. This necessitates an unbiased selection of samples that are mapped to the same output value by the model to be evaluated, and the output distribution for generalization. In contrast to prior works which perturb attributes to expand samples beyond designated test sets, our study is different in two key aspects: first, the cited papers did not investigate the distribution of representative samples mapped to the same output in the entire input space; second, they did not recognize the implication of the output distribution for evaluating model performance.
>   2. Our approach is different from a perturb-and-evaluate framework, wherein sample generation is the major objective. On the contrary, our emphasis lies in the identification of major biases that models may potentially harbor,  thereby gaining an understanding of the tasks that the models might perform inadequately. This transcends the narrow confines of scrutinizing isolated properties or attributes in other existing works.
>   3. Our approach goes beyond established methods and settings: we are the first to advocate **the consideration of output distribution of a ML model** for assessing its performance and gaining a comprehensive understanding and quantification of the predominant biases inherent in a model. As the intrinsic complexity of our new method far surpasses conventional 'perturb-and-evaluate' methods, we made a deliberate choice to compromise scalability in the current manuscript for the sake of establishing a proof-of-principle of the capability and implication of our framework. Nevertheless, the methodology is generic, generalizable, and scalable to other scenarios when computational resources permit.
>   4. All the four cited papers used other generators to generate samples for evaluating a model. On the contrary, we used a sampler to sample the model to be evaluated. Sampling is transparent with convergence estimates, but other generators are still considered as black boxes. Given the inherently unknown biases in models such as CLIP[3], StyleGAN[2], LLM[4], and generative models [1,4], utilizing other models to evaluate a model (as explained in the Discussions section in our manuscript) carries the risk of yielding unfair and potentially incorrect conclusions. Our method brings the focus back to the model to be tested, tasking it with generating samples by itself for scrutiny, rather than relying on external agents such as human or other models to come up with testing data. An additional benefit is that this approach offers a novel framework for estimating errors **in the entire input space** when comparing different models.

---

> > ### Author Response · Authors · 2023-11-20
> > **Could you please check if our revision will make our work meet the ICLR bar?**
> >
> > Thank you for your valuable comments and feedback! We added experiments for **representative datasets** (CIFAR100 and CIFAR10) with **multi-class classification**. In the related work section, we compared the papers with ours in the manuscript and rebuttal in detail. We also replied to the scalability of larger models. Could you please check and see if our revision will make our work meet the ICLR bar? Please let us know if you have any questions. Really appreciate that!

---

> > > ### Comment · Reviewer_BbXK · 2023-11-22
> > >
> > > Learning a sufficiently capable generative model is a well-recognized way to model the input data distribution. It is more intriguing to apply fair samplers on the generative models to sample representative subpopulations. However, I appreciate the effort of this detailed response and will raise the rating from reject to borderline reject.

---

> > > > ### Author Response · Authors · 2023-11-22
> > > >
> > > > Thank you for your comment. Would you mind elaborating on "apply fair samplers on the generative models to sample representative subpopulations?" It seems it is referring to limiting the data to input data distribution.
> > > >
> > > > We agree that it will be an interesting application (as future work). However, our current problem setting and description of the solutions have been clear that the current settings on the pre-defined dataset won't provide as good an understanding of models compared to all possible inputs. One of the reasons is overconfident predictions where previous work [1] elaborated on uninformative samples as well. More importantly, we have demonstrated models are hard to be trusted to model the distribution as we might expected, even for the generative models and other models that we believe to be able to evaluate other models.
> > > >
> > > > [1] Anh Nguyen, Jason Yosinski, and Jeff Clune. Deep neural networks are easily fooled: High confi- dence predictions for unrecognizable images. In Proceedings of the IEEE Conference on Com- puter Vision and Pattern Recognition, pp. 427–436, 2015.

---

> > > > > ### Comment · Reviewer_BbXK · 2023-11-22
> > > > >
> > > > > To be specific, I would be interested in the evidence of the claim that "models are hard to be trusted to model the distribution as we might expected, even for the generative models". Modern large-scale generative models (i.e., Foundation Models) have been empirically shown capable of modeling the data distribution.

---

> > > > > > ### Author Response · Authors · 2023-11-22
> > > > > >
> > > > > > Hello. Would you mind listing the papers of the (foundation) models that "have been empirically shown capable of modeling the data distribution?" It will be valuable for us to include a discussion on them. It'd be better if they don't use (another-)model-based metrics such as FID which we have evidence it is not as reliable to deal with the samples that are different from its training distribution. Really appreciate that!

---

> ### Author Response · Authors · 2023-11-22
>
> Thank you for your reply. Our evaluation framework focuses on the settings where we cannot guarantee any input distribution when the model is deployed. This is extremely important for model safety in the open-world setting where we do not have prior knowledge of what usage cases we might encounter that have not yet been tested. Assuming the data to be tested coming from some input distribution modeled by a generative model limits the generalization of our conclusion. Even if the task is specified and task-specific data distribution is modeled by a generative model, our framework won't change. What is going to change is the sampler that needs to sample according to that distribution, but this is future work not the focus of our framework.
>
> Lastly, previous work [1] has shown even (modern?) generative models like VAE, pixelCNN or flow-based models are not (completely) capable of modeling data distribution, “density learned by flow-based models, VAEs, and PixelCNNs cannot distinguish images of common objects such as dogs, trucks, and horses (i.e. CIFAR-10) from those of house numbers (i.e. SVHN), assigning a higher likelihood to the latter when the model is trained on the former.”
>
> [1] Nalisnick, E., Matsukawa, A., Teh, Y. W., Gorur, D., & Lakshminarayanan, B. (2018). Do deep generative models know what they don’t know?. arXiv preprint arXiv:1810.09136.

---

### Author Response · Authors · 2023-11-16
**Common concerns on representative dataset and multi-class classification**

We thank the reviewers for their constructive comments and suggestions. A common concern from multiple reviewers seems to be the scalability of our framework to different neural network models or larger inputs. Here we first provide a general discussion addressing the scalability topic, our individual responses to reviewers’ questions would then follow.

+ (BbXK, XWBZ, HAeb, 8n4h) Scaling to multi-class classification:
  1. The classes in a multi-class classification problem employ one-hot encoding to represent  the ground truth labels. The output of the model is obtained by applying a log-softmax function to the prediction vector, with each dimension ranging from (-∞, 0]. This formulation allows for the sampling of each output dimension individually, akin to the approach employed in binary classification and generative model scenarios.

+ (BbXK, XWBZ, HAeb, 8n4h) Scaling to larger inputs:
  1. The primary objective of the experimental section of OmniInput is to evaluate and rank the model performance across diverse architectures and training methodologies. Our findings underscore the efficacy of OmniInput in discerning substantial performance differences among various training methods and architectures with simple dataset, whereas the accuracy only reveals the same performance. The observations and explanations provided in the discussion  section about p(x) not learned by the classifiers generalize to other modern deep classifiers for larger inputs. To substantiate our findings, we have presented results on CIFAR-10, where the conclusions drawn about the classifiers persist. Moreover, we provide results on an transformer-based NLP model with a decent number of tokens (up 66) to verify that our method can scale to applications such as NLP model and dataset, and thus our method is universally applicable. Additional results pertaining to multi-class classification for both CIFAR-10 and Cifar-100 are presented in Section I in Appendix (in Supplementary Material); the conclusion remains consistent (refer to the following sections). Notably, the observed performance on classifiers, as assessed through precision, recall, and representative samples, maintains a consistent trend across varying input sizes for both large and small inputs.

  2. Alternatively, sampling smaller inputs and up-sampling to larger inputs for output distribution is a pragmatic approach for assessing performance differences when novel training methods or architectures are introduced. Examining the entire input space even just for smaller inputs, as opposed to relying on alternative models or predefined datasets, can yield valuable insights into the model's performance.

  3. Furthermore, parallel sampling techniques for output distribution have proven to be an efficient means of enhancing performance, as elucidated in the related work presented in [1].

+ To verify the scalability of multi-classes and larger inputs (CIFAR10 and CIFAR100), we add a new Section I in Appendix with detailed discussion of the output distribution and representative samples.

[1] Weitang Liu, Yi-Zhuang You, Ying Wai Li, Jingbo Shang “Gradient-based Wang-Landau Algorithm: A Novel Sampler for Output Distribution of Neural Networks over the Input Space.” Proceedings of the 40th International Conference on Machine Learning

---

### Meta-Review · Area_Chair_Yu89 · 2023-12-22

**Metareview:**

OmniInput evaluates the precision-recall of a model by estimating representative samples across its entire input space, annotating these samples, and then scoring the outputs. Its coverage of the inputs and outputs is derived from the model itself, by relying on sampling powered by the recent WGL sampler, and not supported by particular samples either selected in a dataset or generated from a model. In this way OmniInput potentially offers a more comprehensive, and indeed universal, evaluation of the model across the entire input space. The experiments are not as universal, in that they are restricted to toy datasets and simple classification problems, and indeed focus on binary classification (although this was expanded in the revision, the scope is still restricted to small classification problems in input size and output size in the number of classes). The experiments do show that in these toy settings the analysis by explicit evaluation of the test set (such as AUPR with different OOD sets) does differ from that yielded by the OmniInput evaluation of sampled inputs.

Four expert reviewers are borderline with ratings of borderline/accept (XWBZ, HAeb) and borderline/reject (BbXK, 8n4h) with combined expertise on model evaluation/interpretation/benchmarking, out-of-distribution detection, and training for generalization/robustness. The authors provided specific rebuttals to each review along with a general response. Three reviewers acknowledge the response and engage in discussion and two raise their scores from borderline/reject to borderline/accept (HAeb) and from reject to borderline reject (BbXK).

The argument for acceptance is in the difference of the proposed approach w.r.t. existing methods for the selection, generation, or adversarial examination of inputs and outputs.
OmniInput relies on the model itself (+ a few human annotations) in a way that is more purely about the model than existing approaches that require selection of the test data and generative modeling or simulation to replace or supplement such data. While there is some empirical evaluation of the approach, this is limited compared to other work on model evaluation (as evidenced by many of the works cited in the "open-world model evaluation" section of the related work). Therefore the value is in the methodological novelty of the approach and the potential for further use, and not the technical novelty nor its empirical comprehensiveness.

The argument for rejection is that the method is essentially that of WGL (Liu et al. 2023), the experiments are limited to a toy scale and few methods, and the outlook for scaling to larger inputs (== resolutions, lengths), models (== larger convnets, ViTs), and outputs (== more classes) is restricted by speculative improvement on samplers. Furthermore, as a different class of evaluation, the validation of the method and its possible sensitivities are important to establish its potential for impact in the community. More specifically, this includes further characterizing the dependence of the method on human annotations, especially as the annotations are required of quite different sorts of inputs than humans normally rate (that is, natural inputs). More precisely, the "Evaluation Effort, Efficiency and Human Annotation Ambiguity" section should be expanded and experimentally compared to evaluation on datasets of equal numbers of samples to those needed by OmnitInput.

The AC sides with rejection. Although there is value in different approaches, and while the revision and discussion have improved the submission with ratings rising from rejection to borderline, the reviews remain far from a consensus for acceptance and the AC does not see justification for championing the present edition of this work. That said, the AC does encourage the authors to incorporate feedback and resubmit especially with broader empirical validation across models, comparison with baselines, and confirmation of assertions about human annotations and their amount vs. data-centric evaluation. While scaling to larger datasets and models would of course be more compelling, as shown by the consistent review feedback, a more thoroughly-supported edition of the present submission at the same scope should still be informative to the community as a proof of concept.

Note: The AC acknowledges the polite and detailed comment from the authors regarding unanswered questions during the author-reviewer discussion phase. While every thread could not be concluded during the discussion, the AC considered each point, and assures the authors that ungrounded comments re: foundational models (BbXK) have been removed from the decision.

**Justification For Why Not Higher Score:**

- The technical and empirical contributions are not sufficiently developed. On the technical front, the GWL sampler is much of the method (XWBZ) and it is taken wholly from prior work. On the empirical front, the submission does not compare to baselines beyond the elementary calculation of AUPR (BbXK), and the experiments are limited to the extremely simple datasets (MNIST variants, part of CIFAR-10) which are not a comprehensive experimental framework (BbXK, XWBZ, HAeb). While allowances should be made for work that is exploratory and different, there is still an important matter of degree, and reviewers agree on the need to see OmniInput perform on more data.
- There is missing related work on conditioning evaluation on a given model to better measure its predictions. For two examples, see 1. 3db: A framework for debugging computer vision models (NeurIPS'2022) and 2. Identifying model weakness with adversarial examiner (AAAI 2020). These works generate and select and select data to more thoroughly evaluate a model, and the data for evaluation is a function of the model predictions. While the proposed approach is aligned with the generative model evaluation cited in the submission, it is also related to these threads, and potential readers would be better informed by a discussion of both directions.
- There is an open question about how sensitive OmniInput is to the annotation of its representative inputs by human raters. Although Sec. 5 compares human annotations with machine scores from generative model evaluation, this is not the same as further studying the reliance on human annotations. Likewise the preliminary evaluation of different human annotators and amounts of annotation (Figures 3 & 4) needs more support on other datasets, more annotators than only three as could be done by a human study, and with comparison to data-centric evaluation given an equal number of samples. The assertion of footnote 1 that OmniInput is less dependent than data-centric methods is a question for experiment that is worthy of empirical effort.

**Justification For Why Not Lower Score:**

N/A

---

### Decision · Program_Chairs · 2024-01-16

Reject